# High Brightness Diode Laser Based on V-Shaped External Cavity and Beam-Waist Splitting Polarization Combining

**Yufei Zhao [1], Cunzhu Tong [2],\*, Zhipeng Wei [1],\*, Jian Feng [2]** and **Lijie Wang [2]**

1   Key Laboratory of High Power Semiconductor Lasers, School of Physics, Changchun University of Science and Technology, Changchun 130022, China
2   State Key Laboratory of Luminescence and Applications, Changchun Institute of Optics, Fine Mechanics and Physics, Chinese Academy of Sciences, Changchun 130033, China
\*   Correspondence: tongcz@ciomp.ac.cn (C.T.); weizp@cust.edu.cn (Z.W.); Tel.: +86-0431-86176349 (C.T.); +86-0431-85583390 (Z.W.)

**Abstract:** A beam combining method to improve the brightness of diode lasers is proposed based on a V-shaped external cavity spectral beam and beam-waist splitting polarization combination. This design has the outstanding advantages of improving the beam quality, brightness, and versatility of the diode laser. Specifically, an output power over 16W with $M^2$ factors of $1.79 \times 3.92$ (Beam Parameter Product BPP = $0.55 \times 1.22$ mm mrad) at 40 A in the fast and slow axis is demonstrated for a commercial standard cm-bar. Furthermore, the slow axis $M^2$ of the combined laser is improved by 56% compared with that of a single emitter. Additionally, the brightness of 262 MW·cm$^{-2}$·sr$^{-1}$, 136% higher than that of spectral beam combining without using beam-waist splitting polarization, was realized.

**Keywords:** diode laser; external cavity; beam combining; high beam quality and brightness





## 1. Introduction

Diode lasers with the advantages of small size and high efficiency have been widely used in many applications, such as material processing, pumping of solid state and fiber lasers, security, defense, and communication [1–10]. However, further applications of diode lasers are limited by the low optical power and brightness of a single emitter. To improve the power and brightness, it is necessary to make use of beam combining technologies. Beam combining technology generally consists of coherent beam combining (CBC) and incoherent beam combining [11–14]. The applications of CBC are limited owing to the heavy burdens of its phase control system and low efficiency. Particularly, spectral beam combining (SBC) can realize power scaling with good beam quality close to that of a single emitter [15,16]. In SBC systems, a diode array works in an external cavity which locks the emitters at neighboring wavelengths, which are then spatially combined with optical elements. The external cavity usually consists of a Fourier lens, a diffractive optical element (diffraction grating usually), and a coated mirror to employed as output coupler (OC). By using this method, the high power and high beam quality diode lasers have been demonstrated, and an optical power of 4 KW with a brightness of 2500 MW·cm$^{-2}$·sr$^{-1}$ was reported in 2015 [17–22].

However, due to the poor beam quality of the single emitter in the slow axis limited by the inherent disadvantage of the broad waveguide, the brightness cannot be further improved by SBC. Plenty of attempts to improve the beam quality of emitters have been presented such as micro structuring, loss tailoring, slab-coupled optical waveguide lasers (SCOWLs), and tapered diodes [23–25]. SBC based on SCOWLs and tapered diode lasers were proven in 2005 and 2011 [26–29]. SBC with improved emitters gives better beam quality and higher brightness with a complicated production process and low efficiency at the same time. Off-axis SBC is a potential combining technology based on broad diode

laser array, and its beam quality exceeds that of a single emitter in the slow axis [30–32]. In off-axis SBC, the laser mode feedback to the cavity is often controlled by filters. Although off-axis SBC with filters can improve the brightness, its applications are limited owing to the complex system and low efficiency. The V-shaped external cavity spectral beam combining (VSBC), a new type of off-axis SBC with good beam quality and simple structure was demonstrated in 2018. In VSBC, the OC was replaced by a highly reflective mirror (HR mirror) to realize mode control and beam combining. By this method, a high beam quality ($M^2$ = 2.31 × 3.76), and high brightness of 122 MWcm$^{-2}$ sr$^{-1}$ was realized on a commercial diode laser bar [33]. Another way to improve the brightness is the beam-waist splitting polarization combining (BSPC) proved in 2022. A laser with large waist and purity polarization was split into two loops and combined by a polarization beam combiner (PBS). BSPC shows the advantages of simple structure, high efficiency, and universality. The brightness of a broad diode laser and a commercial diode array were improved by 80% [34].

In this paper, we exploit a beam combining system to promote the brightness of diode lasers based on both V-shaped external cavity spectral beam and beam-waist splitting polarization combining (VBSPC). A commercial standard cm-bar was combined by VBSPC.

In Section 1 we introduce the application of diodes. Some methods and attempts for improving the output power and brightness of diodes were introduced and discussed. The principle and experimental of VBSPC were introduced in Section 2 of this paper. VBSPC shows the advantages of simple structure and the versatility of a diode laser. All the materials of VBSPC are commercial products. In Section 3, we discussed the optical and electrical properties of VBSPC and compared them with a traditional SBC and diode array at free running. By this method, the beam quality and brightness were improved significantly with high efficiency. A laser that has a beam parameter product of approximately 1.22 mm mrad in the slow axis direction was demonstrated. This BPP is better than 7.5 mm mrad by about 6.1 times, which was reported by Witte, U. et al. in 2016 [35]. The advantages and the results at 40A of VBSPC are shown in Section 4.

## 2. Materials and Methods

The schematic diagram of VSBC is shown in Figure 1. The VSBC setup includes a diode laser array (anti-reflection coated in the front facet), a fast-axis collimation lens (FAC), a beam transform system (BTS), a slow-axis collimation lens (SAC), a Fourier lens (TFL), a half-wave plate (HWP), a reflection grating, and an HR mirror. The beams from the diode array are collimated in the fast axis, rotated 90°, and collimated in the slow axis by the SAC. The position error of the emitters due to the smile effect in the horizontal direction is converted to that in the vertical direction. The effect of smile, crosstalk of each emitter, and the divergence of the slow axis are reduced. The BTS increases the stability of the cavity significantly [36]. The HWP is inserted to correct the polarization between lasers from the diode array and the grating. The diode array and the reflection grating are respectively placed on the front and rear focal planes of the transform lens with the focal length $f_T$. The position of each array element was transformed into the angle of incidence on the grating. The HR mirror supplies partial feedback with an off-axis reflecting angle α shown in Figure 1b. Thus, all emitters of the array are forced to work at a specific wavelength in the external cavity structure formed between the back facet of the laser array and the HR mirror. All beams from the diode array are spatially overlapped on the grating at different incident angles with different wavelengths; they diffract at the same angle, and are perpendicular to the HR mirror in the direction of the laser. In principle, the beam quality of VSBC goes beyond the single emitter and traditional SBC. The wavelength of each emitter can be calculated by $\lambda_i = \Lambda(\sin\theta_i + \sin\theta_d)$. $\lambda_i$ is the wavelength of each emitter, the $\theta_i$ and $\theta_d$ are the incident angle in the grating and the diffraction angle, respectively. The linewidth of VSBC $\Delta\lambda$ can be calculated by $\Delta\lambda \approx d * \Lambda * \cos\theta_L / f_T$. The $d$ is the lateral dimension of the array, $\Lambda$ is the grating period, and the $\theta_L$ is the Littrow angle.

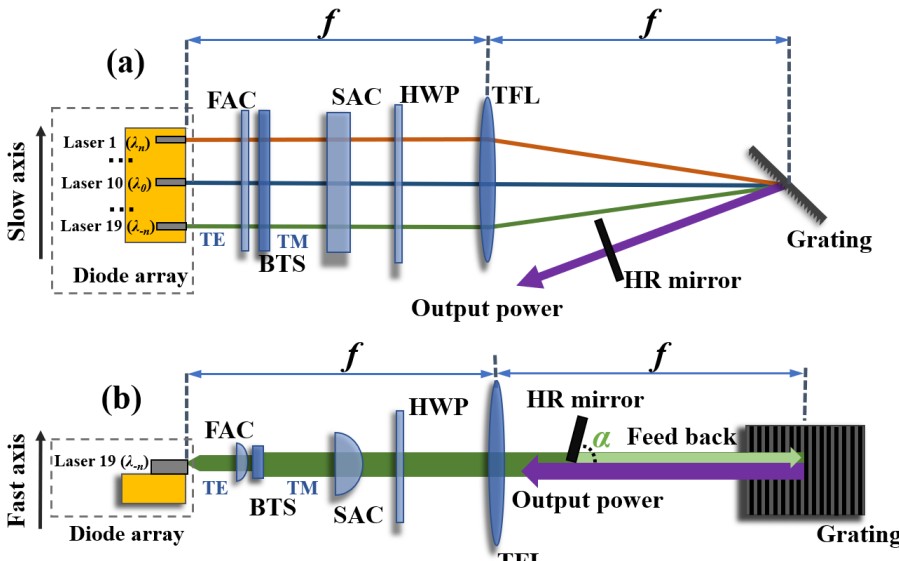

**Figure 1.** Setup for VSBC in the slow axis direction (**a**), VSBC in the fast axis direction (**b**).

A standard centimeter BAL bar with 19 emitters working at 940 nm is employed in our experimental setup. All the emitters are coated with a reflectivity of R < 5% on the front facet. The parameters of the diode laser array are shown in Table 1. The BTS and the FAC with focal lengths of 0.286 mm used are commercial products from LIMO. The SAC is a cylindrical lens with a focal length of 7 mm. The focal length of the transform lens is 350 mm. All optical components mentioned above have an anti-reflection coating of R < 0.5%. The diffraction grating is 1800 lines/mm with a first-order diffraction efficiency of 91% at 940 nm. From the measured result, we found that the grating brings a significant fraction of the loss about 10%. The HR mirror is a K9 mirror coated with Au.

**Table 1.** The Parameters of the 940 nm Diode Laser Array.

| Parameters | Value |
|:---:|:---:|
| Wavelength/nm | 940 |
| Chip width/mm | 10 |
| Cavity length/mm | 1.5 |
| Number of emitters | 19 |
| Emitter pitch/µm | 500 |
| Emitter width/µm | 100 |
| Cavity length/mm | 1.5 |
| The thickness of the waveguide/µm | 1 |
| $\theta_{\text{fast}}$ 95%power content/(°) | 45 |
| $\theta_{\text{slow}}$ 95%power content/(°) | 7 |
| Polarization | TE |

To improve the beam quality further, a BSPC was employed after the VSBC. The schematic of the experimental setup of BSPC is shown in Figure 2. BSPC includes the laser combined VSBC, a precisely cut prism, three HR mirrors, an HWP, and a beam combiner (PBC). The laser from the array combined by VSBC was split into two loops equally by a precisely cut prism along the beam waist in the slow axis. The polarization of one loop was changed by the HWP. The two loops with different polarization were combined by the (PBC) using three HR mirrors. The beam quality will be improved by 50% in principle because that the beam waist of the combined laser by BSPC is reduced to half of that original. BSPC is suitable for the laser from VSBC.

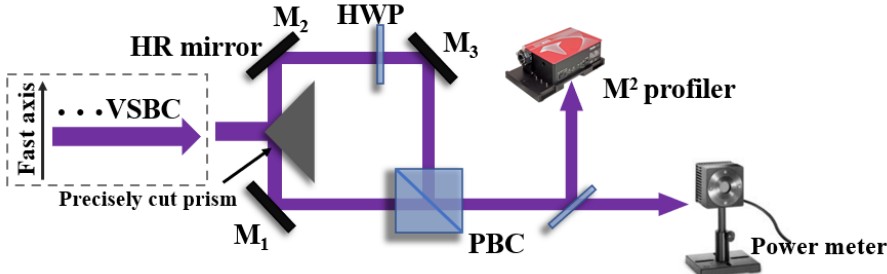

**Figure 2.** Schematic diagram of BSPC setup.

In our experimental setup, the three HR mirrors ($M_1$, $M_2$, $M_3$) with Au coating of R > 97.8% at 980 nm are made of K9 glass. The HWP used is a zero-order half-wave plate with a transmission of 98.5%. The precisely cut prism is from Thorlabs (MRAK25). The size of the PBC is 20 mm × 20 mm × 20 mm, and the combining efficiency of is 96.5%.

## 3. Results

The power curves of the array under free running (red), SBC at an OC with reflectivity of 33% (blue), VSBC with maximum output power (yellow), and VBSPC (green) are shown in Figure 3. With the coolant temperature of 25 °C, the power of the four devices was measured by Ophir FL500A. Too small power feedback leads to an unstable combining, and large overlapping reduces the combined power for VSBC. The output power of the laser bar is 30.2 W when it is under free running at 40 A, and the electro-optic conversion efficiency is 51.2%. The power of VSBC is 17.4 W at the same driven current and the combining efficiency is 57.6%. With the same current, the power of SBC at an OC with a reflectivity of 33% is 16.7 W and the beam combining efficiency is about 55%. The power of the VBSPC is 16.2 W, 7% lower than that of VSBC and the beam combining efficiency is 53.6%. The power loss of VBSPC compared with VSBC is caused by additional optical elements. The combining efficiency of VBSPC is 1.3% lower than that of SBC at an OC with a reflectivity of 33%. The reasons for the combining efficiency of the four devices lower than 71% [33] are (1) the laser array we used in this paper was coated with a reflectivity about 5% higher than 0.5%, (2) more optical components that bring energy loss were employed in VBSPC. From the measured result, we found that the grating brings a significant fraction of the loss, about 10%. VBSPC shows a combining efficiency close to that of SBC.

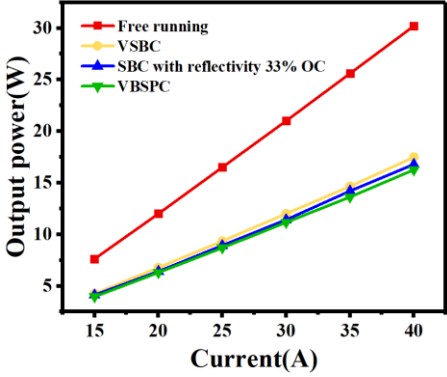

**Figure 3.** The power–current characteristics of free running (red), SBC at an OC with reflectivity of 33% (blue), VSBC with maximum output power (yellow), and VBSPC (green).

The $M^2$ factors in the fast axis ($M^2_y$) and slow axis ($M^2_x$) of a single emitter constituting the laser bar (blue), the SBC at an OC with reflectivity of 33% (green), VBSPC (red), and VSBC (yellow) are shown in Figure 4a,b from 15 A to 40 A. The $M^2$ measurement system is from Thorlabs (M2MS). The $M^2_y$ of the four devices are very close in the range from 1.4 to 1.88 and increase slowly. At 40 A, the $M^2_y$ of SBC, VSBC, and VBSPC are higher

than that of a single emitter. Maybe the reason is the imperfect overlapping of beams on the grating caused by spherical aberration of the transform leans. With the increase in driving current, the $M^2_x$ values of the four devices increased very fast. The high driving current leads to the increase of high-order mode operation [25] and the effect of lateral carrier accumulation also brings the rise of $M^2_x$ [37]. At 40A, the $M^2_x$ of the single emitter, SBC, VSBC, and VBSPC increased to 8.97, 9.33, 7.4, and 3.92. $M^2_x$ of VBSPC shows an improvement of 56%, 52%, and 57% better than that of the single emitter, VSBC, and SBC, respectively. The noticeable improvement of $M^2_x$ is attributed to the mode selection of VSBC and the narrowed waist by BSPC. VBSPC improves beam quality significantly with an efficiency close to that of SBC.

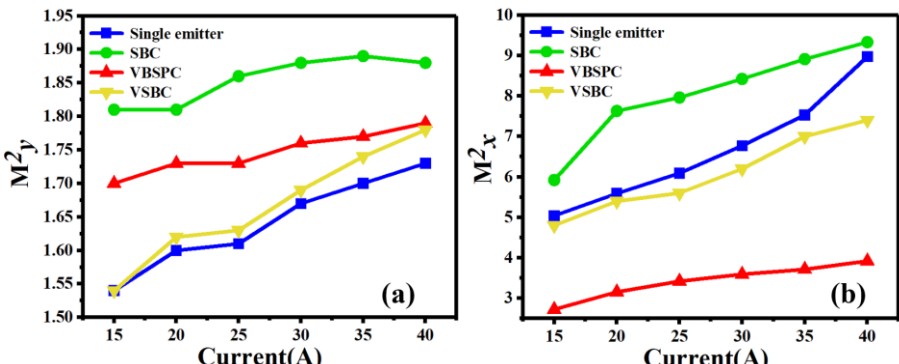

**Figure 4.** The dependence of $M^2_y$ (**a**), $M^2_x$ (**b**) factors of VBSPC (red), SBC at an OC with reflectivity of 33% (green), VSBC (yellow), and single emitter (blue) on current.

Another important parameter for laser is brightness, which is defined as follows:

$$B = \frac{P}{\lambda^2 M_x^2 M_y^2} \tag{1}$$

In Equation (1), the $P$ and the $\lambda$ are the output power and the center wavelength of the laser (940 nm), respectively. The brightness of SBC (green), SBC with BSPC (yellow), and VBSPC (red) with the increase in the driving current are shown in Figure 5. The brightness of SBC and VBSPC initially increases and reaches 110 MW·cm$^{-2}$·sr$^{-1}$, 204.1 MW·cm$^{-2}$·sr$^{-1}$, and 262 MW·cm$^{-2}$·sr$^{-1}$ from 15 A to 40 A. The brightness of VBSPC is about 2.36 times and 1.28 times higher than that of SBC and VSBC. The attainment brightness of VBSPC is owing to the excellent $M^2$ factors and high combining efficiency.

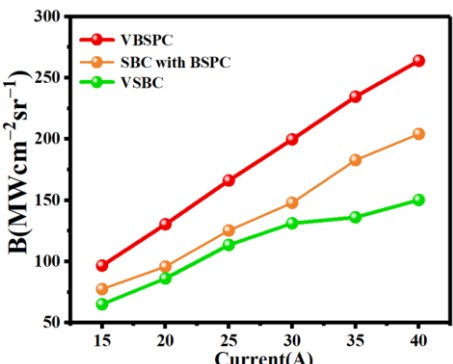

**Figure 5.** The brightness of SBC (green), SBC with BSPC (yellow), and VBSPC (red) with respect to the injection current.

We also measured the spectra of VBSPC at 40 A and show it in Figure 6. As can be seen, the lasing spectrum of VSBC consists of nineteen peaks. The peaks are clear with no split, which means that there is no crosstalk between the emitters. The measured

wavelength spread $\Delta\lambda$ was 11.14 nm, which agrees with the value of 11.18 nm calculated using the equation.

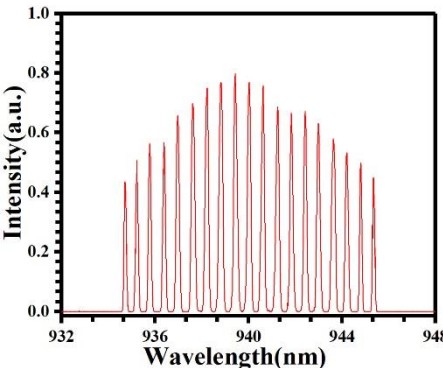

**Figure 6.** The wavelength of VBSPC at 40 A.

## 4. Discussion

In summary, a beam combination technique to improve the brightness of diode lasers based on both V-shaped external cavity spectral beam combining and beam-waist splitting polarization combining was proposed and demonstrated. The VBSPC setup shows the high beam combining efficiency approaching that of SBC, good beam quality exceeding a single emitter, and high brightness. A commercial standard cm-bar was combined by this system. A 16.2 W laser with $M^2$ factors of $1.79 \times 3.92$ in the fast and slow axis was demonstrated at 40 A. The brightness of the combined laser is as high as $262 \text{ MW·cm}^{-2}\text{·sr}^{-1}$, 136% higher than that of spectral beam combining without using beam-waist splitting polarization ($110 \text{ MW·cm}^{-2}\text{·sr}^{-1}$) at 40 A. We believe that the VBSPC will make contributions to beam combining techniques.

## 5. Patents

C.Z.Tong, F.Y.Sun, Y.F.Zhao, S.L.Shu, L.J.Wang, "Spectral beam combined laser system and method", US patent 10,333,265.

C.Z.Tong, Y.F.Zhao, F.Y.Sun, S.L.Shu, L.Wang, S.C.Tian, L.J.Wang, "Laser beam combining system", US patent 10,768,434.

**Author Contributions:** Methodology, Y.Z.; validation, C.T. and Y.Z.; investigation, Y.Z. and L.W.; data curation, J.F.; writing—review and editing, Y.Z. and C.T.; supervision, C.T. and Z.W.; project administration, C.T. and Y.Z. All authors have read and agreed to the published version of the manuscript.

**Funding:** National Natural Science Foundation of China (62025506).

**Institutional Review Board Statement:** Not applicable.

**Informed Consent Statement:** Not applicable.

**Data Availability Statement:** Not applicable.

**Conflicts of Interest:** The authors declare no conflict of interest.

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
