# Peer review of "High Brightness Diode Laser Based on V-Shaped External Cavity and Beam-Waist Splitting Polarization Combining"

_applsci, doi:10.3390/app13042125_

Round 1

Reviewer 1 Report

The presented study is a logical continuation of the studies carried out by the authors [27, 28].

I expected to see a comparison of three approaches to brightness increase: VSBC, BSPC and VBSPC, however, only VBSPC and VSBC are compared. Why is it so? It can be seen that the beam quality does not increase much compared to a single VSBC.

Specify the laser array pumping mode: pulsed or continuous? If there is a pulsed mode, then what is the pulse duration and pulse repetition rate?

Please, name the element that introduces the main optical loss in the system: is it BTS?

Please, indicate the initial divergence of a single emitter from the laser array, and also, if possible, the thickness of the waveguide in the laser heterostructure.

Please, specify the width of the emitting aperture of the laser array. It is necessary to give a description of the laser array: emitters’ widths, how the lateral current confinement of the active stripes is organized?

Why the power of a free running laser array is so low: 30W at 40A?

What is the sensitivity of the beam quality to the rotation of the laser array around the optical axis and to the laser array smile?

In the introduction, please add references to studies on microstructuring the active stripe with single-mode lasers, an effective approach to a lateral brightness increase.

Please, explain the purpose of placing a diffraction grating at the focus of the TFL, since a diffraction grating is known to operate effectively with a collimated beam only. The grating operates correctly for a collimated beam when the entire grating is illuminated. How the resolving power of the diffraction grating changes in the case of a beam focused on it? Please, add the grating name given by the manufacturer.

Please, add polarization direction markings to the rays in Figure 1. How the diffraction grating is oriented with respect to the polarization of the incident beam. Does BTS change the polarization direction?

Authors state that «slow axis M2 of the combined laser is improved by 56% compared with that of a 17 single emitter». This may simply be due to vignetting or diaphragming in the optical system. How many times is the total power of the system greater than the output power of a single emitter?

Please, give a more detailed description of how “external cavity forces the emitters work at neighboring wavelengths” (line 33).

The BTS with a focal length of 0.286 mm could not be found in the site of manufacturer (https://www.focuslight.com/search?cid1=50&cid2=57&cid3=66&goods=553). Please, provide a product code.

«The laser array is collimated in fast axis, rotated 90° by the BTS». Describe why it is important.

To estimate the contribution of PBC, please, add to Figure 4 a comparison with the VSBC arrangement.

Combining efficiency in [https://doi.org/10.1364/OE.26.014058] was 71%. Here the combining efficiency is 57.6%. What causes this drop?

Is the output power of a laser array the same for both cases of SBC and VBSPC arrangements in the current range from 15A to 40 A?

How the brightness of 262[MW*cm^−2*sr^−1] is calculated for a 16.2W laser with M2 factors of 1.79 ×3.92?

The power sensor Ophir FL500A that was used is a BDFL500A-BB-50 Ophir power sensor?

Please, advise me a program that was used to draw the figure 1.

I propose to remove the phrase “combined by this system” from the sentence on line 16, since it does not make sense.

Line 42: “plate coupled optical waveguide lasers (SCOWL)” replace with “slab-coupled optical waveguide lasers (SCOWLs)”.

Line 68: add “beam” after “laser array”.

«262 MW·cm−2·sr−1, 136% higher than traditional spectral beam combining». I propose to write something like “spectral beam combining without using beam-waist splitting polarization” instead of the phrase “traditional spectral beam combining”.

Reviewer 2 Report

The authors present an experimental setup of a beam combining technique that aims to display a laser beam of a higher power. The authors report a brightness performance of 262 MW/cm2/sr. The idea is interesting. But, I could recommend the manuscript for publication if the following modifications are done:

1-The authors use the words diode laser and semiconductor lasers. If there is a difference between the two, the authors must clarify. Otherwise, this needs to be uniform.

2- The paper deals with the performance of diode lasers. But, in the setups of Figs.1 and 2, one cannot identify a laser diode. It is then difficult to follow.

3- Figures 1 and 2 need more detailed explanations of the functioning principle instead of just citing different components as is the case in the first and the second paragraphs of section 2.

4- In the last paragraph of the introduction, are the authors talking about the results of someone else or their results?

5- The structure of the paper is not given.

6- The best results of the authors are achieved at 40A:

a-       Is that value not too large since semiconductor lasers are known to consume low energy?

b-       The plots of figures 3 to 5 are increasing curves. Why have the authors stopped at 40A ?   What is the physical meaning of that increasing?

c-       In the conclusion, the authors reported that “…  136% higher than that of traditional spectral beam combining”. The value obtained by tradition beam combining must be presented and referenced as well as the operating current.   

7- More applications of semiconductor lasers must be included in the introduction. See for instance “Modulation of distributed feedback (DFB) laser diode with the autonomous Chua’s circuit: Theory and experiment, Opt. Laser Tech, https://doi.org/10.1016/j.optlastec.2017.09.042;  Complex photonics: dynamics and applications of delay-coupled semiconductors lasers, Rev. Mod.Phys, https://doi.org/10.1103/RevModPhys.85.421; D. Patil, « Semiconductor Laser Diode Technology and Applications », InTech, 2012.

Reviewer 3 Report

        In the submission by Yu-fei Zhao et. al., the manuscript with the title “High brightness diode laser based on V-shaped external-cavity and beam-waist splitting polarization combining” demonstrated a method to obtain the high-brightness diode lasers. In the manuscript, the diode array is combined by V-shaped external cavity beam combining and beam-waist splitting polarization combining. In the V-shaped external cavity beam combining, the output coupler is replaced by a high reflectivity mirror, and the mode of the output laser was selected. As a result, in beam-waist splitting polarization combining, the combined beam waist can be reduced by 50% in principle. With this method, a laser with beam quality exceeding that of a single emitter was achieved, and then the brightness of the diode array is improved significantly.

      In my option, this manuscript contains some interesting results, and I recommend its publication. However, I still have two suggestions to improve the readability of the manuscript in the following:

     (1) Some quantitative results about plate-coupled optical waveguide lasers (SCOWL) and Off-axis SBC should be given in the abstract;

     (2) The explanation “The power-current characteristics of free running (red line), SBC at an OC with reflectivity of 33% (blue line), VSBC with maxim output power (yellow line), and VSBC with BSPC (green line).” of Figure 3 does not match the text in the manuscript, the authors should check it again and give a correction. 

Author Response

Response to Reviewer 3 Comments

In the submission by Yu-fei Zhao et. al., the manuscript with the title “High brightness diode laser based on V-shaped external-cavity and beam-waist splitting polarization combining” demonstrated a method to obtain the high-brightness diode lasers. In the manuscript, the diode array is combined by V-shaped external cavity beam combining and beam-waist splitting polarization combining. In the V-shaped external cavity beam combining, the output coupler is replaced by a high reflectivity mirror, and the mode of the output laser was selected. As a result, in beam-waist splitting polarization combining, the combined beam waist can be reduced by 50% in principle. With this method, a laser with beam quality exceeding that of a single emitter was achieved, and then the brightness of the diode array is improved significantly.

      In my option, this manuscript contains some interesting results, and I recommend its publication. However, I still have two suggestions to improve the readability of the manuscript in the following:

  • Some quantitative results about plate-coupled optical waveguide lasers (SCOWL) and Off-axis SBC should be given in the abstract;

      Response 1: I have added quantitative results about plate-coupled optical      waveguide lasers (SCOWL) and Off-axis SBC in the introduction.

     (2) The explanation “The power-current characteristics of free running (red line), SBC at an OC with reflectivity of 33% (blue line), VSBC with maxim output power (yellow line), and VSBC with BSPC (green line).” of Figure 3 does not match the text in the manuscript, the authors should check it again and give a correction.

Response 1: I have checked and corrected it.

Round 2

Reviewer 1 Report

The proposed VBSPC system is characterized by large dimensions, the spectum width of the entire system is expected to be very large. The introduction contains few examples of efficient systems.

Please, provide information on the spectrum width of the proposed VBSPC setup at 40A.

In the intoduction, please compare your results with such studies as [U. Witte, F. Schneider, M. Traub, D. Hoffmann, S. Drovs, T. Brand, and A. Unger, "kW-class direct diode laser for sheet metal cutting based on DWDM of pump modules by use of ultra-steep dielectric filters," Opt. Express 24, 22917-22929 (2016)] and discuss on brightness of 2500 MW/(cm2 sr), optical power 4 kW from a 100 µm (NA ~0.08) with an efficiency of 44% [Recent progress on high-brightness kW- class direct diode lasers,” in Proc.of IEEE Conf. on High Power Diode Lasers and Systems (IEEE, 2015), pp. 29–30]. Please, add the beam parameter product values that are necessary for comparison.

In the introduction, please add references on microstructuring the active stripe, for example [Platz, R., Erbert, G., Pittroff, W., Malchus, M., Vogel, K., & Tränkle, G. (2013). 400 µm stripe lasers for high-power fiber coupled pump modules. High Power Laser Science and Engineering, 1(1), 60-67. https://doi.org/10.1017/hpl.2012.1].

Please, add the “half-wave plate” also to the side-view (Fig.1(b)).

Your answer “Response 16” is very inforamtive and useful. Please, add it into the text.

Line 44: “plate coupled” replace with “slab-coupled”.

Line 92: “emits” replace with “emitter”

Reviewer 2 Report

Dear editor, I am almost satisfied with the answers given by the authors.

But, my last concerns are:

1- In the last paragraph of the introduction, are the authors talking about the results of someone else or their results? Then put these references at the indicated place.

2- The structure of the paper is not given. The structure means the plan of the paper that should appear at the end of the introduction. For instance, Section 2 deals with. In Section 3, we …..
